# Non-Invasive Monitoring of Increased Fibrotic Tissue and Hyaluronan Deposition in the Tumor Microenvironment in the Advanced Stages of Pancreatic Ductal Adenocarcinoma

**DOI:** 10.3390/cancers14040999

**Published:** 2022-02-16

**Authors:** Ravneet Vohra, Yak-Nam Wang, Helena Son, Stephanie Totten, Akshit Arora, Adam Maxwell, Donghoon Lee

**Affiliations:** 1Department of Radiology, University of Washington, Seattle, WA 98109, USA; vohrar@uw.edu; 2Applied Physics Laboratory, University of Washington, Seattle, WA 98105, USA; ynwang@uw.edu; 3Division of Gastroenterology, University of Washington, Seattle, WA 98104, USA; helenas@uw.edu (H.S.); sitotten@uw.edu (S.T.); 4Department of Art and Sciences, University of Washington, Seattle, WA 98195, USA; aeora44@uw.edu; 5Department of Urology, University of Washington School of Medicine, Seattle, WA 98195, USA; amax38@uw.edu

**Keywords:** pancreatic ductal adenocarcinoma (PDAC), multi-parametric magnetic resonance imaging (mp-MRI), apparent diffusion coefficient (ADC), amide proton transfer (APT), chemical exchange saturation transfer (CEST), glycosaminoglycan (Gag)

## Abstract

**Simple Summary:**

Pancreatic ductal adenocarcinoma (PDAC) is a deadly disease with a poor prognosis. A better understanding of the tumor microenvironment may help better treat the disease. Magnetic resonance imaging may be a great tool for monitoring the tumor microenvironment at different stages of tumor evolution. Here, we used multi-parametric magnetic resonance imaging techniques to monitor underlying pathophysiologic processes during the advanced stages of tumor development and correlated with histologic measurements.

**Abstract:**

Pancreatic ductal adenocarcinomas are characterized by a complex and robust tumor microenvironment (TME) consisting of fibrotic tissue, excessive levels of hyaluronan (HA), and immune cells. We utilized quantitative multi-parametric magnetic resonance imaging (mp-MRI) methods at 14 Tesla in a genetically engineered KPC (*Kras^LSL-G12D/+^*, *Trp53^LSL-R172H/+^*, *Cre)* mouse model to assess the complex TME in advanced stages of tumor development. The whole tumor, excluding cystic areas, was selected as the region of interest for data analysis and subsequent statistical analysis. Pearson correlation was used for statistical inference. There was a significant correlation between tumor volume and T2 (r = −0.66), magnetization transfer ratio (MTR) (r = 0.60), apparent diffusion coefficient (ADC) (r = 0.48), and Glycosaminoglycan-chemical exchange saturation transfer (GagCEST) (r = 0.51). A subset of mice was randomly selected for histological analysis. There were positive correlations between tumor volume and fibrosis (0.92), and HA (r = 0.76); GagCEST and HA (r = 0.81); and MTR and CD31 (r = 0.48). We found a negative correlation between ADC low-b (perfusion) and Ki67 (r = −0.82). Strong correlations between mp-MRI and histology results suggest that mp-MRI can be used as a non-invasive tool to monitor the tumor microenvironment.

## 1. Introduction

Pancreatic cancer is the fourth leading cause of cancer-related deaths in the USA [1]. The higher death rate in pancreatic cancer is partly attributable to the complex nature of the tumor microenvironment (TME), which is difficult to monitor and target using novel therapeutic drugs. Pancreatic ductal adenocarcinoma (PDAC), unlike other tumors, displays robust stroma and a dense extracellular matrix (ECM), which is often termed as desmoplasia [2]. The desmoplasia continuously evolves during the disease progression to include higher numbers of stromal fibroblasts, immune cells, excessive deposition of ECM, and a complex mixture of proteoglycans and glycosaminoglycans (Gag) [3]. PDAC tumor cells produce exceedingly high concentrations of Gag in the tumor interstitium [2,4], which can lead to exceedingly high interstitial fluid pressure (IFP). An increase in IFP leads to the collapse of the tumor vasculature making it even more difficult to treat. Therefore, a clear understanding of TME may help in targeting specific underlying pathophysiological changes and effective delivery of therapeutic drugs to the tumors.

In pre-clinical and clinical trials, the invasive measures required to evaluate TME, thus far have failed to provide time-dependent changes in the PDAC. Therefore, there is a dire need to develop non-invasive techniques to monitor disease progression as well as study the therapeutic effects of existing and novel chemotherapeutic agents. Magnetic resonance imaging (MRI) has proven to be successful in (1) monitoring the tumor micro-environment and (2) providing high resolution and high contrast images of the tissue. Methods such as quantitative T1 and T2 relaxation times [5,6,7,8], magnetization transfer (MTR) [9], Apparent diffusion coefficient (ADC) [10] are well-established examples, and more recently, chemical exchange saturation transfer (CEST) MRI has been applied in the characterization of cancer [11,12,13].

A single MR parameter may not be sufficient to provide the complete picture of the complex nature of TME. Multi-parametric MRI (mp-MRI) is beneficial because we can use different sequences to monitor the underlying pathological processes. In order to monitor the complex and everchanging TME, it is imperative that we utilize multi-parametric-MRI (mp-MRI) thereby facilitating the development and testing of appropriate therapeutic regimens. Indeed, we have previously demonstrated effectiveness of mp-MRI based approach and found that T2 and MTR were sensitive to increase in fibrotic tissue accumulation [7,8] during the early stages of tumor development. In this study, we expanded mp-MRI acquisitions at 14 Tesla (T) utilizing the T1, T2, MTR, diffusion weighted (DW), and CEST imaging sequences and correlative appropriate histological measurements to monitor the TME in the advanced stages of PDAC in *KPC* mice.

## 2. Materials and Methods

### 2.1. Mouse Strains

The study was conducted with the approval from our institutional animal care and use committee (IACUC) of the University of Washington. All mice were housed in a specific pathogen free, controlled environment (14 h/10 h light/dark cycle, 73.5 ± 5 °F) with ad libitum access to tap water and chow. The animal protocol was designed to minimize pain or discomfort to the animals included in our study. We used the *Kras^LSL-G12D/+^*, *Trp53^LSL-R172H/+^*, *Cre* (*KPC*) genetic PDAC mouse model. *KPC* animals conditionally express endogenous mutant *Kras* and point mutant *Trp53* alleles, spontaneously develop PDAC, and closely mimic the pathophysiology and molecular progression of the human disease. *KPC* mice (*n* = 24; age (mean ± S.D) = 6.0 ± 2.5 months; body weight = 27.5 ± 4.9 gms) were enrolled in the study when they had a tumor mass > 250 mm^3^, confirmed by ultrasound imaging and MR imaging. All mice were euthanized at the terminal time point and the tumor and associated pancreatic tissue was excised and prepared for histological evaluation.

### 2.2. MRI Protocol

MRI experiments were performed on a 14T Bruker Avance 600 MHz/89 mm wide-bore vertical MR spectrometer (Bruker Corp., Billerica, MA, USA). A birdcage coil (inner diameter 25 mm) was used to image the animal mounted on a cradle with a respiratory monitoring probe. General anesthesia was induced by 1.5% isoflurane mixed with oxygen. Once the animal was anaesthetized, the animal was secured to the custom-built cradle. The coil was then inserted vertically into the scanner, which was kept at 30 °C to maintain body temperature of the animal. The entire MR data acquisition took 50–60 min during which the animals were continuously monitored for respiratory rate. Following the MR acquisition, mice were removed from the coil and allowed to recover on a custom-built warm waterbed before returning to their respective cages.

#### 2.2.1. Anatomical Images

The multi-slice MRI protocol, covering the whole tumor, started with fat-suppressed T_1_ weighted coronal images (repetition time (TR) = 2000 ms; echo time (TE) = 5.49 ms; number of averages (NA) = 1, field of view (FOV) = 30 × 30 mm; matrix size = 128 × 256; yielding spatial resolution of 0.234 × 0.117 mm/pixel) for anatomical reference. Subsequently, 20 axial images were acquired to which were used to calculate tumor volume.

#### 2.2.2. T1 Weighted Imaging

Multiple images using rapid acquisition with refocused echoes were acquired using following parameters: TE = 9.66 ms, TR = 5500, 3000, 1500, 1000, 385.8 ms, NA = 1, FOV = 30 × 30 mm, matrix size = 256 × 128; yielding spatial resolution of 0.117 × 0.234 mm/pixel. The data acquisition time was approximately 9 min.

#### 2.2.3. T2 Weighted Imaging

Multiple spin-echo data were acquired in coronal orientation covering the area from liver to kidneys. The quantitative T2 maps were generated using a multi-slice multi echo sequence, with fat signal suppressed, utilizing following parameters: TR = 4000 ms; TE = 12 echoes equally spaced from 6.28 ms to 75.4 ms; NA = 1; FOV = 30 × 30 mm; matrix size = 256 × 128 yielding spatial resolution of 0.117 × 0.234 mm/pixel. To cover the entire abdominal region, 10 contiguous slices were acquired without any inter slice gap. All the images were gated to respiration of the animal. The data acquisition time was approximately 6 min.

#### 2.2.4. Magnetization Transfer (MT)

MT ratios (MTR) were acquired using a gradient echo sequence (TR/TE = 625/2 ms, flip angle = 30°) with an off-resonance frequency of 7000 Hz and a saturation pulse block pulse shape, 50 ms width, and 10 µT amplitude. A series of 10 images were acquired with FOV = 30 × 30 mm, matrix size = 256 × 256 yielding spatial resolution of 0.117 × 0.117 mm/pixel. The acquisition time for data acquisition was approximately 3 min.

#### 2.2.5. Diffusion Weighted Images (DWI)

A pulsed gradient spin echo diffusion measurement (pulse duration = 3.0 ms and diffusion time = 7.46 ms) was performed to acquire series of 10 slices using following parameters: TR = 2500 ms; TE = 17.7 ms; NA = 1; FOV = 30 × 30 mm; matrix size = 128 × 128 yielding spatial resolution of 0.234 × 0.234 mm/pixel. Diffusion weighted measurements were acquired with 8 different b values (0, 30, 60, 100, 150, 200, 300, 500 s/mm^2^). The data acquisition time was approximately 10 min.

#### 2.2.6. Chemical Exchange Transfer Saturation (CEST) Imaging

On a single 1 mm slice, delineating tumor, amide proton transfer (APT) imaging was performed with respiratory gating using small animal monitoring device (SA instruments, Inc., Stony Brook, NY, USA). Gradient-echo images were acquired following a pre-saturation pulse (continuous-wave block pulse, B1 = 0.5 μT, duration = 2 s) which was applied at 25 frequency offsets from −360 Hz to 360 Hz with an interval of 0.5 ppm to estimate a center frequency shift. Other imaging parameters were as follows: TR/TE = 2200/7 ms, FOV = 30 × 30 mm, matrix size = 128 × 128, flip angle = 180°, and number of excitations = 8. For saturation, an off-resonance RF pulse was applied for 3 s at a power level of 2 μT, TR/TE = 5000/7 ms, matrix = 128 × 128, FOV = 30 × 30 mm, slice thickness = 1 mm, single slice). Finally, a control image with the saturation offset at 300 ppm was also acquired. Total acquisition time for each animal was approximately 19 min.

### 2.3. Tissue Preparation and Histology

Mice were sacrificed, and the tumors were gently removed and immediately frozen in OCT for histological analysis. Three serial 5 μm sections were cut every 1 mm through the entire tumor (CM1950, Leica, Bannockburn, IL, USA). Tissue between the section steps was collected for biochemical analysis. Samples taken for biochemical analysis were evaluated for connective/fibrotic tissue accumulation (Masson’s trichrome), and hyaluronan (HA) using HA binding protein (HABP, Millipore Sigma, Burlington, MA, USA) [7]. Primary antibodies and reagents used were CD31 1:100 (Abcam ab28364, Waltham, MA, USA), and Ki67 1:150 (Abcam ab16667, Waltham, MA, USA).

### 2.4. Image Analysis

#### 2.4.1. MR Image Analysis

All raw MR images were processed using Image-J software (http://imagej.nih.gov/ij/ (accessed on 19 September 2021)), to measure mean values of the different tumors. Anatomical images were used to measure tumor volume. Tumor volume was measured using HOROS software. T1 and T2 maps: Maps were generated using T_1_ and T2 weighted images. MTR maps: The MTR was measured using the following ratio: (SI_0_ − SI_s_/SI_0_), where SI_0_ represents the tissue signal intensity without saturation pulse applied while SI_s_ represents the tissue signal intensity with saturation pulse. Diffusion maps: Diffusion weighted MR signal decay was analyzed using mono-exponential model: S_b_/S_0_ = exp.(−b∙ADC). Where S_b_ is the MRI signal intensity with diffusion weighting b, S_0_ is the non-diffusion-weighted signal intensity and ADC is the apparent diffusion coefficient. Three lowest b values of 0, 30 and 60 s/mm^2^ were used to calculate perfusion component (or pseudo-diffusion) whereas rest of the 5 b values of 100, 150, 200, 300 and 500 s/mm^2^ were used to calculate tissue diffusivity component. CEST images were quantified for APT using the following equation: [S_sat_ (−3.5 ppm) − S_sat_ (3.5 ppm)]/S_0_ where S_sat_ and S_0_ are the water signal intensities measured with and without saturation pulse. For the glycosaminoglycan spectrum (GagCEST), maps were generated using a similar calculation for the 0.5, 1.0 and 1.5 ppm frequency shifts (Appendix A).

#### 2.4.2. Histological Image Analysis

All histological image analysis was done using a custom script written for MATLAB (2020b, The Mathworks, Natick, MA, USA). A region of interest encompassing the entire tissue cross-section was first manually selected. Tissue vs. background area was determined by conversion of the color image to grayscale and intensity thresholding. For colorimetric analysis, the image was converted to L*A*B* color space (a three-dimensional method to define colors like RGB), and pixels were categorized by the built-in k-means clustering function *kmeans* [14] with cluster number 3 and using 3 replicates. Each pixel was further quantified by its root-mean-squared difference of A* and B* from each cluster center. Selection of tissue with the smallest distance to the highest-order (highest mean of A*, B*) cluster identified collagenous tissue, which appears bright blue against other tissue stained typically purple or red in a Masson’s trichrome section [15]. From these values, the fractional area of collagen within the tissue was calculated (Appendix A).

## 3. Results

### 3.1. T1, T2 and MTR vs. Tumor Volume

T2 (r = −0.65, *p* = 0.0006) and MTR (r = 0.52, *p* = 0.009) were significantly correlated T1 (r = 0.14, *p* = 0.52) did not demonstrate any correlation with tumor volume in advanced stages of tumor development (Figure 1).

### 3.2. Apparent Diffusion Coefficient (ADC) vs. Tumor Volume

ADC maps were generated using bi-exponential model of intravoxel incoherent motion (IVIM), which involves both perfusion and diffusion components [16,17]. ADC values significantly correlated with increase in tumor volume (r = 0.49, *p* = 0.014). At low-b values, moderate correlation was found between tumor volume and perfusion (r = −0.40, *p* = 0.02). Similarly, at high b values, a moderate correlation was found between the tumor volume and diffusion component (r = 0.57, *p* = 0.004) (Figure 2).

### 3.3. Chemical Exchange Saturation Transfer (CEST) Imaging vs. Tumor Volume

Amide proton transfer (APT) imaging is one subset of CEST imaging, referring to chemical exchange between protons of free tissue water (bulk water) and amide groups (-NH) of endogenous proteins. It has been reported that such protons are more abundant in the tumor microenvironment than in healthy tissues [18,19]. APT CEST values increased with increase in tumor volume (r = 0.28, *p* = 0.21), whereas there was moderate correlation between tumor volume and GagCEST values (r = 0.51, *p* = 0.02) (Figure 3).

### 3.4. Correlative Histological Measures

MR data was correlated with histological measures in a subset on KPC mice (*n* = 6). Fibrotic tissue accumulation was significantly correlated with tumor volume (r = 0.92), and MTR (%) (r = 0.78). There was an inverse correlation between fibrotic tissue accumulation and T2 (r = −0.61) and ADC low b (r = −0.70) (Figure 4). HA content was significantly correlated with the tumor volume (r = 0.76). Furthermore, there was a significant correlation between HA content and GagCEST (r = 0.81). Additionally, there was a moderate correlation between CD31 and MTR (%) (r = 0.48). Finally, there was a significant inverse correlation between ADC-low b and Ki67 (r = −0.82) (Figure 5).

## 4. Discussion

The current study evaluated the TME in the advanced stages of pancreatic cancer using mp-MRI in the *KPC* mouse model. The results from the present study are as follows: (1) T2 demonstrated a significant negative correlation with an increase in tumor size, (2) MTR (%) demonstrated a significant positive correlation with increase in tumor size, (3) Using the IVIM model, tumor perfusion decreased significantly whereas diffusion increased in advance stages of PDAC, (4) Finally, there was an increase in GagCEST and APT CEST with an increase in tumor volume in advanced stages of cancer.

The measurement of any MR parameter in isolation discounts the dynamic nature of TME. However, the use of mp-MRI enables the evaluation of multiple parameters to give a more representative picture of the TME. Methods such as quantitative T1, T2, ADC and MTR are well established methods used to characterize tumor progression [6,10,20,21]. T1 and T2 relaxation times of water molecules in tissue have been demonstrated as sensitive indicators of tumor progression as well as responses to different therapeutic agents [22,23]. T1 and T2 times are sensitive to factors such as the amount of (1) water in the extracellular space and (2) protein in the water [24]. Additionally, T2 measures are also sensitive to deposition of dense collagenous tissue in the extra-cellular space [25]. Similarly, in the present study we have demonstrated a decrease in T2 with increase in tumor volume in the advanced stages of the tumor development. Additionally, in the previous study, we established the reduction in T2 was due to increase in fibrotic tissue deposition [8]. There was a moderate but significant correlation between T2 and fibrotic tissue deposition in both the studies.

MTR values are sensitive to fibrotic tissue deposition and have been used to study liver fibrosis and PDAC [20,26,27]. A study by Farr et al. demonstrated that the *KPC* mouse model displays significantly higher fibrotic tissue deposition than other mouse models [27]. Furthermore, a previous study by our group demonstrated that MTR is significantly correlated with the tumor volume and fibrotic tissue accumulation once the tumor volume crossed the threshold value of 250 mm^3^ [8]. Similarly, in the present study we have demonstrated that MTR is significantly correlated with the tumor volume in the advanced stages of tumor development.

The ADC values can reflect the Brownian motion of water molecules in tissue extracellular and intercellular space and in tissue microcirculation [17,28,29]. ADC derived from diffusion weighted imaging (DWI) values are usually affected by cellular density, fiber content, the degree of blood supply and physical movement [30,31,32,33]. The decrease of ADC values is mainly dependent on the increase of cellular density and fiber content, which limits the Brownian motion of water molecules. Variable values of the ADC in PDAC have been reported, with lower and higher ADC values of PDAC compared with normal pancreatic tissue. For example, Yoshikawa et al. reported that pancreatic cancer had a higher ADC value than normal pancreas [34] while some studies showed no significant difference in ADC values between pancreatic tumor and normal pancreatic parenchyma [35,36,37,38]. However, an increased body of evidence have shown that ADC values of PDAC are much lower than those of normal pancreas [10,31,39,40]. Interestingly, in the present study we have demonstrated a significant increase in ADC in the advanced stages of PDAC tumor. One could reasonably argue that in the presence of dense stroma, the diffusion values in the tissue must decrease. However, due to the complex nature of TME in PDAC, it demonstrates an inverse relationship. One of the reasons that could be attributed to increase in ADC is the presence of high concentration of HA in the tumor [2,3,4]. Furthermore, we have demonstrated a significant positive correlation between tumor volume and GagCESTin the advanced stages of PDAC.

Based on the intravoxel incoherent motion (IVIM) model, DWI with sufficient b values enables to separately reflect microcapillary perfusion and tissue diffusivity [17,28,29,41,42]. Intravoxel incoherent motion diffusion weighted imaging (IVIM-DWI) can simultaneously provide information about tumor perfusion and diffusion characteristics without using contrast medium. The signal from blood flow is rapidly attenuated at low b values (b < 100–150 s/mm^2^), whereas higher b values are required to suppress the perfusion contribution. Indeed, quantitative parameters derived from IVIM-DWI have been increasingly used to diagnose and differentiate pancreatic lesions [37,42,43,44]. Results from the present study demonstrated a moderate but negative correlation between tumor volume and ADC low b (perfusion), which is similar to previously published results from our group [8]. Furthermore, significant positive correlation was seen between tumor volume and ADC-high b values, suggesting increase in diffusivity. While these capabilities are helpful, there is further potential for MRI contrast to represent a specific biomarker and to obtain quantitative estimates of these biomarkers.

CEST contrast is induced by the effects of a saturation pulse applied at the resonance frequency of protons which are in exchange with freely moving water [45]. The exchange interaction results in a reduction of signal in the free water, which is detectable as a decrease in intensity of the MRI image. CEST contrast helps explore the changes in tumor metabolism and cellular density associated with tumor growth and response to therapy [46,47]. Altered CEST contrast can represent changes in the local chemical environment of specific metabolites and has been linked to altered cellular metabolism pathways and apoptosis [48,49]. Indeed, CEST has been used to examine cancer in the brain [50] and the breast [51,52] leading to the detection of increased amide proton transfer (APT) in tumor because of differences in pH and protein content inside cells. The increased APT signal found in tumors has been attributed to an increase in free protein concentration in malignant cells [19,50,53]. While APT has been studied in depth in oncology applications, hydroxyl exchange has been a largely unexplored area, particularly in the pancreatic tumors, where glycosaminoglycan (GAG) content could be an important indicator of tissue status. A previously published study has presented an increase in HA accumulation in PDAC tumors [3,4]. Likewise, our group demonstrated targeted depletion of HA after PEGylated recombinant human hyaluronidase (PEGPH20) and Gemcitabine [54]. Our study has a few limitations that need to be acknowledged. First, we did not quantify interstitial fluid pressure (IFP) which has been correlated with the increase in HA accumulation [2]. Second, we did not compare the MR and histological values with age-matched control pancreatic tissue.

It is important that preclinical studies be performed in such a way that the results can be translated to clinical studies. The KPC model is a valuable resource to study the tumor microenvironment in PDAC. Additionally, the use of a non-invasive technique such as MRI could be a useful tool for the evaluation of novel therapeutic agents to treat PDAC. Furthermore, mp-MRI can provide indispensable information regarding the tumor microenvironment.

## 5. Conclusions

We found that there is increased deposition of fibrotic tissue, and hyaluronic acid in KPC mice. Furthermore, we believe that the quantitative mp-MRI measures used in this study have potential role in monitoring these subtle changes in the tumor microenvironment at the later stages of tumor development.

## Figures and Tables

**Figure 1 cancers-14-00999-f001:**
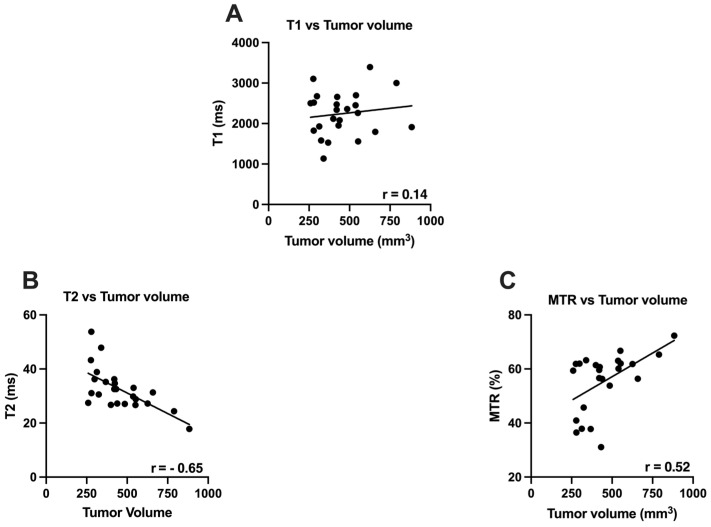
Relationship between Tumor volume and MR parameters in advanced stages of PDAC. (**A**) showing relationship between Tumor volume and T1 (ms); (**B**) showing relationship between Tumor Volume and T2 (ms); (**C**) showing relationship between Tumor volume and MTR (%).

**Figure 2 cancers-14-00999-f002:**
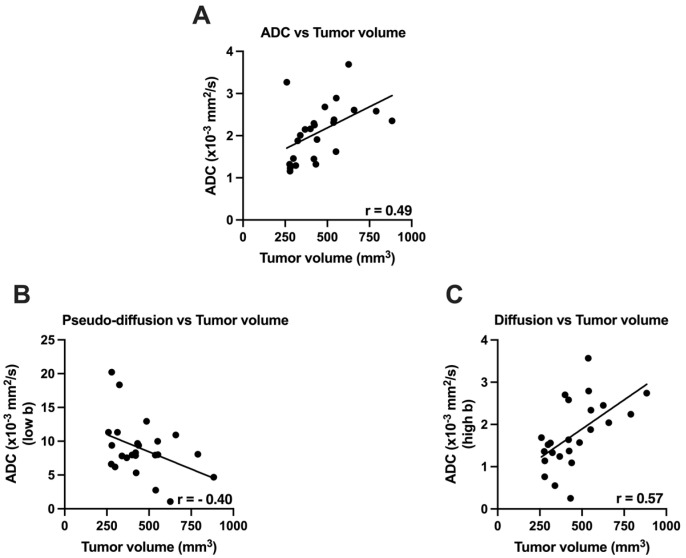
Relationship between Tumor volume and ADC in advanced stages of PDAC. (**A**) showing relationship between Tumor volume and ADC (mm^2^/s); (**B**) showing relationship between Tumor Volume and ADC low b values (pseudo-diffusion); (**C**) showing relationship between Tumor volume and ADC (high b values).

**Figure 3 cancers-14-00999-f003:**
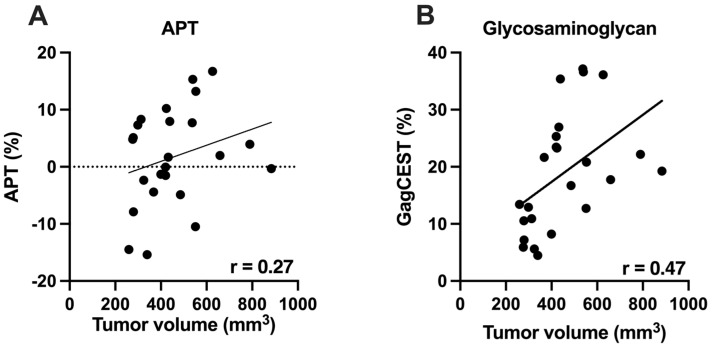
Relationship between Tumor volume and CEST parameters in the advanced stages of PDAC. (**A**) showing relationship between Tumor volume and Amide proton transfer (APT%); (**B**) showing relationship between Tumor Volume and Glycosaminoglycans (GagCEST).

**Figure 4 cancers-14-00999-f004:**
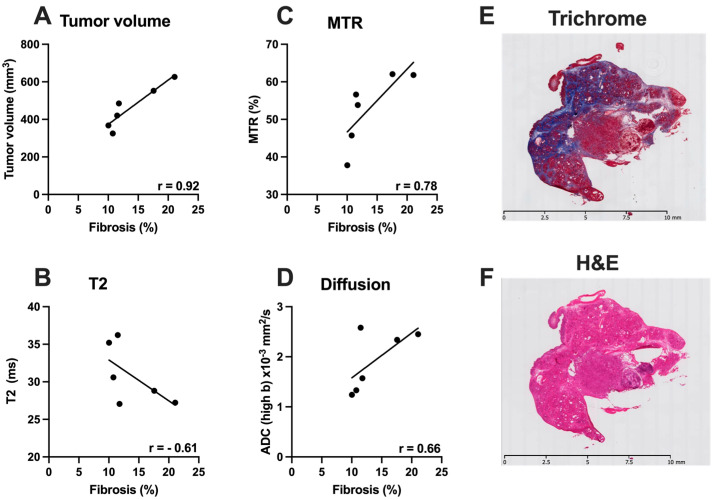
Relationship between MR and histological parameters in the advanced stages of PDAC. (**A**) showing relationship between Tumor volume (mm^3^) and Fibrosis (%); (**B**) showing relationship between T2 (ms) and Fibrosis (%); (**C**) relationship between MTR (%) and Fibrosis (%); (**D**) ADC low b (×10^−3^ mm^2^/s) and Fibrosis (%); (**E**,**F**) Representative images of Masson’s Trichrome and H&E stained tumor.

**Figure 5 cancers-14-00999-f005:**
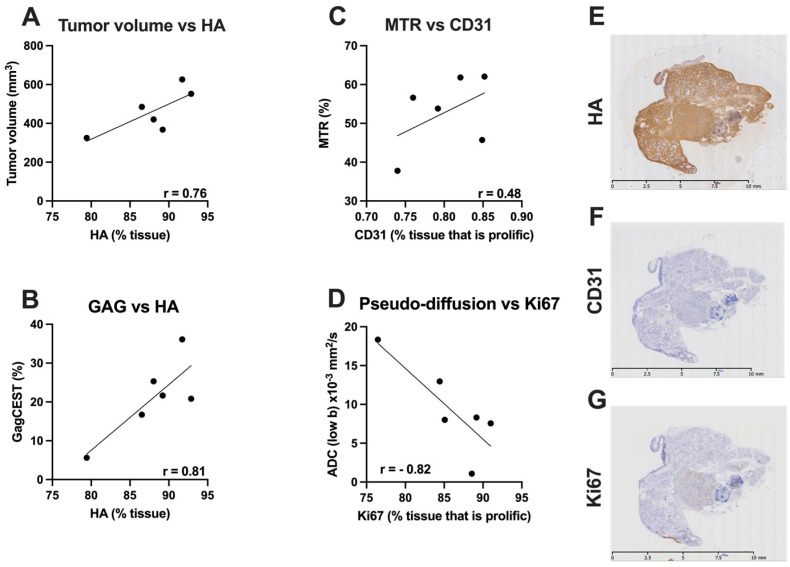
Relationship between MR and histological parameters in the advanced stages of PDAC. (**A**) showing relationship between Tumor volume and Hyaluronan (HA%); (**B**) showing relationship between GagCEST and HA (%); (**C**) relationship between MTR (%) and CD31; (**D**) ADC low b and Ki67. (**E**–**G**) Representative images of HA, CD31 and Ki67 stained slides.

## Data Availability

The data presented in the current study are available on reasonable request from the corresponding author.

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
