# Peer review of "Non-Invasive Monitoring of Increased Fibrotic Tissue and Hyaluronan Deposition in the Tumor Microenvironment in the Advanced Stages of Pancreatic Ductal Adenocarcinoma"

_cancers, 2022, doi:10.3390/cancers14040999_

Round 1

Reviewer 1 Report

In the current study the authors utilized quantitative multi-parametric magnetic resonance imaging     techniques to assess Pancreatic Ductal Adenocarcinoma progression in KPC mice in an effort to develop non invasive techniques; the authors have also correlated the MRI data to appropriate histological measurements.The study is well organized, the authors provide a well-justified introduction, and their experiments are well-designed and systematically presented.The study is quite interesting and probably opens new diagnostic noninvasive methods with clinical relevance in the monitoring appropriate therapeutic regimens.

Comments

  1. What was the average age (± SD) and weight (± SD) of animals at the time of enrollement into the study?
  2. Have the authors estimated connective tissue with Masson’s trichrome  
  3. Are there any limitations in the study?
  4. It would be interesting if the authors mention the clinical perspectives in a separate paragraph
  5.      Add characteristic pictures, i.e of Gag cest contrast map  

Author Response

We are pleased to submit a revised version of our manuscript (ID: cancers-1546934) entitled “Non-invasive monitoring of increased fibrotic tissue and Hyaluronan deposition in the tumor microenvironment in the advanced stages of Pancreatic Ductal Adenocarcinoma.” We thank the reviewer for the thoughtful comments. We have addressed the individual comments below.

1. What was the average age (± SD) and weight (± SD) of animals at the time of enrollement into the study?

We have addressed this comment in line 90.

2. Have the authors estimated connective tissue with Masson’s trichrome  

We have quantified amount of connective tissue using Masson’s trichrome and have addressed this comment in line 155.

3. Are there any limitations in the study?

We have addressed this comment in lines 427-431.

4. It would be interesting if the authors mention the clinical perspectives in a separate paragraph

We have addressed this comment and added a paragraph for clinical perspective from lines 432-437.

5. Add characteristic pictures, i.e of Gag cest contrast map  

We have added colored maps for T1, T2, MTR, ADC, and GagCEST as a supplementary figure (Figure S1).

Reviewer 2 Report

Comments:

Well presented paper and any imaging or analysis monitoring PDAC is useful data towards some understanding of this intractable cancer. 

1) Please check spacing before publication as it alters through manuscript sections.

2) Is the custom script for histological image analysis publicly available? (line 170)

3) The way in which the cluster was chosen and analysis done is not clear from the description (line 169-176). Can an example image be shown and how this was done? How was bias excluded? How was area of tumour chosen or was whole tumour analysed?

4) Figure 1: Very blurred from copy and paste. Please copy and insert / paste as metafile to see if that helps? Graphs have no titles? 

5) Figure 2: Graphs have no titles. As above, very blurred, please fix. 

6) Graphs: Please ensure all graphs are inserted to retain quality and add titles.....

7) Can images have scale bars inserted please? Also magnification details in Figure legends...

Author Response

We are pleased to submit a revised version of our manuscript (ID: cancers-1546934) entitled “Non-invasive monitoring of increased fibrotic tissue and Hyaluronan deposition in the tumor microenvironment in the advanced stages of Pancreatic Ductal Adenocarcinoma.” We thank the reviewer for the thoughtful comments. We have addressed the individual comments below.

1) Please check spacing before publication as it alters through manuscript sections.

Thank you very much for pointing this out. We have checked and made necessary corrections.

2) Is the custom script for histological image analysis publicly available? (line 170)

The custom script for histology was written in house and is not publicly available. However, it can be made available upon request.

3) The way in which the cluster was chosen and analysis done is not clear from the description (line 169-176). Can an example image be shown and how this was done? How was bias excluded? How was area of tumour chosen or was whole tumour analysed?

We further explained how the histological image analysis was performed in lines 179-187. Also, we have uploaded a supplementary figure (Figure S1) demonstrating various steps taken for histology quantification.

4) Figure 1: Very blurred from copy and paste. Please copy and insert / paste as metafile to see if that helps? Graphs have no titles? 

We have updated the figure with high-resolution images.

5) Figure 2: Graphs have no titles. As above, very blurred, please fix. 

We have updated the figures and included the graph titles. 

6) Graphs: Please ensure all graphs are inserted to retain quality and add titles.....

We have updated the figures and included the graph titles. 

7) Can images have scale bars inserted please? Also magnification details in Figure legends...

We have added the scale bars on the histology figures.

Round 2

Reviewer 2 Report

The authors have addressed all comments. I have no further comments and the manuscript has been clarified.